# The β_2_-Subunit (AMOG) of Human Na^+^, K^+^-ATPase Is a Homophilic Adhesion Molecule

**DOI:** 10.3390/ijms23147753

**Published:** 2022-07-14

**Authors:** María Luisa Roldán, Gema Lizbeth Ramírez-Salinas, Marlet Martinez-Archundia, Francisco Cuellar-Perez, Claudia Andrea Vilchis-Nestor, Juan Carlos Cancino-Diaz, Liora Shoshani

**Affiliations:** 1Department of Physiology, Biophysics and Neurosciences, CINVESTAV-IPN, 2508 IPN Ave., San Pedro Zacatenco, Ciudad de México 07360, Mexico; mroldan@fisio.cinvestav.mx (M.L.R.); pacupe_1@hotmail.com (F.C.-P.); cvilchis85@gmail.com (C.A.V.-N.); 2Department of Immunology, Instituto de Investigaciones Biomédicas, Universidad Nacional Autónoma de México (UNAM), Circuito, Mario de La Cueva S/N, C.U., Coyoacán, Ciudad de México 04510, Mexico; gemali86@hotmail.com; 3Laboratorio de Modelado Molecular, Bioinformática y Diseño de Fármacos, Departamento de Posgrado Escuela Superior de Medicina del Instituto Politécnico Nacional, Salvador Díaz Mirón esq. Plan de San Luis S/N, Miguel Hidalgo, Casco de Santo Tomas, Ciudad de México 11340, Mexico; mtmartineza@ipn.mx; 4Departamento de Microbiología, Escuela Nacional de Ciencias Biológicas del Instituto Politécnico Nacional, Salvador Díaz Mirón esq. Plan de San Luis S/N, Miguel Hidalgo, Casco de Santo Tomas, Ciudad de México 11340, Mexico; jcancinod@ipn.mx

**Keywords:** AMOG, Na^+^, K^+^-ATPase β_2_-subunit, cell adhesion molecule (CAM), protein–protein interaction, protein docking, homophilic interaction

## Abstract

The β_2_ subunit of Na^+^, K^+^-ATPase was originally identified as the adhesion molecule on glia (AMOG) that mediates the adhesion of astrocytes to neurons in the central nervous system and that is implicated in the regulation of neurite outgrowth and neuronal migration. While β_1_ isoform have been shown to trans-interact in a species-specific mode with the β_1_ subunit on the epithelial neighboring cell, the β_2_ subunit has been shown to act as a recognition molecule on the glia. Nevertheless, none of the works have identified the binding partner of β_2_ or described its adhesion mechanism. Until now, the interactions pronounced for β_2_/AMOG are heterophilic cis-interactions. In the present report we designed experiments that would clarify whether β_2_ is a cell–cell homophilic adhesion molecule. For this purpose, we performed protein docking analysis, cell–cell aggregation, and protein–protein interaction assays. We observed that the glycosylated extracellular domain of β_2_/AMOG can make an energetically stable trans-interacting dimer. We show that CHO (Chinese Hamster Ovary) fibroblasts transfected with the human β_2_ subunit become more adhesive and make large aggregates. The treatment with Tunicamycin in vivo reduced cell aggregation, suggesting the participation of N-glycans in that process. Protein–protein interaction assay in vivo with MDCK (Madin-Darby canine kidney) or CHO cells expressing a recombinant β_2_ subunit show that the β_2_ subunits on the cell surface of the transfected cell lines interact with each other. Overall, our results suggest that the human β_2_ subunit can form trans-dimers between neighboring cells when expressed in non-astrocytic cells, such as fibroblasts (CHO) and epithelial cells (MDCK).

## 1. Introduction

The Na^+^, K^+^-ATPase (Na^+^-pump) is a transmembrane enzyme responsible for the active transport of Na^+^ and K^+^ ions through the plasma membrane of all animal cells [1]. The Na^+^-pump catalyzes the hydrolysis of ATP as an essential energy source to transport three Na^+^ ions out of the cell and two K^+^ ions into the cell against ion gradients [2]. This important pump is constituted of two indispensable subunits, α and β. The α subunit (110 kDa) contains the binding sites for ATP, Na^+^, K^+^, and sites for inhibitors and activators. All tissue-specific isoforms of the Na^+^, K^+^-ATPase α subunit (α_1_, α_2_, α_3_, and α_4_) share the same catalytic function. The β subunit (35 kDa) is a glycoprotein [3]. Isoforms of the Na^+^-pump β subunit (β_1_, β_2_, and β_3_) do not actuate the catalytic process but are essential to the proper function of the pump. In addition to the essential function of the Na^+^, K^+^-ATPase as an ion pump, recent work has suggested further roles for Na^+^, K^+^-ATPase in signal transduction and cell–cell adhesion. Na^+^, K^+^-ATPase is capable of forming multiple protein–protein complexes. A relatively recent review describes protein–protein interactions for which the α-subunit of Na^+^, K^+^-ATPase serves as an anchoring platform [4]. In the present work, we will focus on protein–protein interactions that involve the β_2_-subunits of Na^+^-K^+^-ATPase.

Epithelial cells are polarized into distinct apical and basolateral plasma membrane domains. Early studies demonstrated that the transporting epithelial phenotype depends on a polarized localization of Na^+^, K^+^-ATPase, specifically at the basolateral membrane domain of the cells [5,6,7]. Although the detailed manner in which the Na^+^, K^+^-ATPase achieves polarized distribution remains unknowable, several pieces of evidence have emerged on the participation of the β subunit in that mechanism [5,8,9,10,11]. In recent years, our laboratory has focused on the β_1_ subunit of the epithelial Na^+^, K^+^-ATPase and its role as a cell adhesion molecule. Initially, we observed that the β subunit of the Na^+^, K^+^-ATPase resembles an adhesion molecule: it has a short cytoplasmic domain, a single transmembrane domain, and a long and glycosylated extracellular domain. Likewise, reports from the group of Schachner [12,13] discovered that the adhesion molecule on glia (AMOG) is the β_2_ isoform of the Na^+^, K^+^-ATPase [14]. Eventually, the structure of the Na^+^-K^+^-ATPase, with its three subunits, has been resolved by X-ray crystallography and revealed that the β_1_-subunit is mostly exposed to the intercellular space [15,16] and shares structural similarities with cell adhesion molecules [17]. In accordance with this feature of the β subunit, we have demonstrated that Chinese hamster ovary cells transfected with the canine β_1_-subunit of Na^+^, K^+^-ATPase (CHO-β_1_) increase their tendency to form aggregates [6]. Using co-cultures of MDCK and CHO cells, we showed that the Na^+^, K^+^-ATPase of MDCK cells was polarized to the lateral border given that the adjacent cell expresses the same species (dog) of β_1_-subunit [6]. We also showed by pull-down assay that the dog β_1_-subunit could specifically bind to the soluble extracellular domain of the β_1_-subunit of the same animal species, and by FRET and Co-IP assays we demonstrated that β_1_-subunits of neighboring epithelial cells interact directly with each other [7]. The fact that the extracellular moieties of β_1_-subunits have a high affinity for each other results in the anchoring of the Na^+^, K^+^-ATPase of both neighboring cells to the lateral border. Recent works identified the interacting amino acids at the interface of β_1_-subunits [18,19].

It is now well established that both β_1_ and β_2_ isoforms function as adhesion molecules in epithelia and the nervous system, respectively. Nevertheless, little is known about the adhesion mechanism of the β_2_/AMOG isoform. It has been shown that β_2_/AMOG functions as a neural recognition molecule mediating neuron-glia interactions and that it promotes neurite outgrowth [12,20]. Subsequent works demonstrated that expression of β2 subunit of Na^+^, K^+^-ATPase in liposomes [13], in glioma cells [21], and L-fibroblasts [20] confer adhesiveness. Nevertheless, none of them identified the ligand molecule for the β_2_-subunit. On the other hand, various works have shown that β2 subunit cis-interacts with membrane proteins such as Basigin/CD147–the accessory subunit of the Mono-carboxylate transporters–and with the Prion-related protein (PrP)–a neural cell adhesion molecule involved in neurite outgrowth, neuronal survival, and synaptic function–which in turn interacts with the AMPA receptor subunit, glutamate receptor 2 (GluR2) [22]. Remarkably, none of the studies show a trans-interaction whether with another β subunit or with another cell adhesion molecule (CAM).

The main structural difference between β_1_ and β_2_ isoforms is their grade of N-glycosylation. Human β_1_ has three sites while β_2_ has seven sites. It has been shown that the glycosylation of β_1_ subunit plays an important role in β1–β1 interaction. N-glycosylation removal by enzymatic, chemical, or mutational methods reduces β1–β1 adhesion [23,24]. Nevertheless, no extracellular lectin has been identified as responsible for this effect [24]. Thus, N-glycans are probably required for stabilizing the amino acid-mediated interactions between the extracellular domains of the two β_1_ subunits. In the case of β_2_, the cis-interaction–at least with Basigin/CD147–has been shown to involve the lectin domain of Basigin/CD147 and the oligomannosidic residues of β_2_ [25]. Nevertheless, little is known about the role of N-glycans of β_2_ in mediating cell–cell trans-interactions. Therefore, we decided to approach some of the open questions related to the β_2_ adhesion mechanism using similar strategies that were used for demonstrating that β_1_ is a homotypic trans-interacting adhesion molecule in epithelial cells. In the present work, we expressed the human β_2_ subunit of Na^+^, K^+^-ATPase in CHO fibroblasts and designed in vitro and in vivo experiments that strongly suggest that the β_2_ subunit is also a homophilic trans-interacting cell adhesion molecule.

## 2. Results

### 2.1. In Silico Prediction of Glycosylated β_2_–β_2_ Trans-Interaction

As a first approach aimed to clarify the adhesive properties of the β_2_ subunit of Na^+^, K^+^-ATPase, we decided to use bioinformatics tools to find out whether β_2_–β_2_ interaction is relevant considering the physicochemical properties of the extracellular domain of the glycosylated protein (EDβ_2_). To begin with, since there is no crystal structure available for ATP1B2, we generated a three-dimensional model. In search of a template for the β_2_ subunit, we selected the β subunit of pig gastric proton pump due to its highest identity with the human β_2_ isoform of Na^+^, K^+^-ATPase (39.82%). Thus, a 3D model of the EDβ_2_ was generated as specified in Methods. Figure 1A illustrates the model obtained by the Swiss Model Program [26]. The corresponding Ramachandran plot depicted in Figure 1B shows that none of the residues are located in the disallowed region. These results suggest that our template-based model is valid. Thereafter, we proceeded to build the dimeric structure of the protein. Protein–protein docking was performed using HDOCK server, which is a specialized program for protein–protein docking analysis [27,28]. HDOCK uses the binding information from the PDB and a hybrid docking algorithm of template-based modeling and free docking [27]. As we were looking for a trans-interaction, we chose conformers in which the membranes are positioned in an opposing (parallel) way. Then, from the list of seven predictive models, model 1, with the higher ranking, and a free binding energy of −274.99 kcal/mol was selected to be illustrated in Figure 1C. The putative binding interface composed of 11 residues in each monomer is depicted in Figure 1D. Interestingly, for both Chains, at least four glycosylation sites of the EDβ_2_ are located in the interface (Asn118, Asn153, Asn159, and Asn197) and their interactions are illustrated in Appendix A.

### 2.2. Generation of Experimental Tools

Encouraged by the in silico results, we designed in vitro and in vivo experiments that would further evaluate the trans-interaction between β_2_ isoforms. Thus, we expressed the human β_2_ subunit of Na^+^, K^+^-ATPase in CHO fibroblasts and in MDCK epithelial cells. We selected these cells, as they do not express that isoform of β subunit. A specific antibody against the human β_2_ subunit does not detect that protein in wild-type CHO fibroblasts or MDCK cells when assayed by immunofluorescence microscopy (IF) (Figure 2A) or by Western blot (WB) (Figure 2B).

Transfection with the cDNA of human β_2_ subunit fused to the yellow fluorescent protein resulted in numerous stable clones of CHO β_2_ YFP and MDCK β_2_-YFP that were positive for β_2_ subunit by both WB analysis and fluorescence microscopy imaging. A representative stable clone of each is shown in Figure 2A. The YFP images were captured by confocal microscopy at 530 nm. As can be noticed, the β_2_ subunit is concentrated in the cell–cell contacts as observed for the β_1_ subunit in wild-type MDCK cells. Cell lysates of both CHO and MDCK β_2_-YFP cells showed an expected molecular weight of approximately 75 kDa when both anti-β_2_ (upper panel) and anti-GFP (lower panel) antibodies have been used (Figure 2B). Moreover, that distribution at cell–cell contacts resembles one of many other cell–cell adhesion molecules. Therefore, we proceeded to study the adhesive properties of CHO β_2_ YFP cells.

### 2.3. Over-Expression of β_2_ YFP Increases Cell Aggregation in a Glycosylation-Dependent Manner

The β_2_ subunit has been shown to function as a cell adhesion and recognition molecule mainly in astrocytes. To test whether the expression of the β_2_ subunit in CHO cells (CHO β_2_ YFP) confers adhesiveness, we used two different adhesion assays.

One is a Dispase I adhesion assay. As described in Methods, a monolayer of CHO β_2_ YFP adhered to the culture dish is treated with Dispase I to detach cell–substrate adhesion. A monolayer of wild-type CHO cells is detached as single cells suspended in the culture medium (Figure 3). On the other hand, epithelial cells firmly attached by adhesion complexes (adherens and tight junctions) detach as a sheet [29]. In the case of CHO β_2_ YFP and CHO β_1_, as shown in Figure 3B,C, the cells detach as big aggregates. These observations suggest that CHO cells overexpressing the β_2_ YFP subunit form a pseudo-epithelial monolayer establishing both cell–substrate and cell–cell interactions. The attached cells are harvested with trypsin and re-suspended as single cells in the second adhesion assay. Then, the cells are allowed to aggregate by incubating at 37 °C overnight in constant agitation. Figure 4 illustrates the results of an aggregation assay performed with CHO β_2_ YFP cells. As controls, we compared CHO wt cells and CHO cells transfected with the canine β_1_ subunit (CHO dog β_1_). The cells were cultured to 100% of confluence, as shown in the confocal microscopy images corresponding to each treatment (Figure 4A). The relative amount of aggregates that have been formed was estimated by flow cytometry.

A representative dot plot is shown for each treatment (Figure 4B). Figure 4C shows the average of aggregates created in three independent experiments. CHO fibroblasts form a small number of aggregates (less than 5%) while the CHO YFP-β_2_ cells form a significantly higher proportion (30%) as well as the CHO dog β_1_ cells (20%). As cell–cell adhesion mediated by β_1_–β_1_ interaction depends on the N-glycosylation of the β_1_ subunits [23,24], we analyzed whether N-glycosylation is involved in the cell-aggregation of CHO YFP-β_2_. As depicted in Figure 4B,C, treatment with Tunicamycin significantly decreases the aggregation of CHO YFP-β_2_ cells. As expected, it also diminished the aggregation of CHO dog β_1_ cells. These experiments show that the adhesive property of β_2_/AMOG of human Na^+^, K^+^-ATPase is conserved when expressed in fibroblasts from hamster ovary (CHO) and that it involves N-glycans. Therefore, we next addressed whether this aggregation could be mediated by β_2_–β_2_ interaction.

### 2.4. Cell Aggregation of CHO β_2_ YFP Decreases When Cocultured with CHO wt Cells

If cell–cell aggregation of CHO β_2_ YFP cells observed by flow cytometry is mediated by the β_2_-subunits of neighboring cells, we expected that diluting the cell suspension with non-transfected CHO cells would reduce the proportion of cell aggregates. On the other hand, if the β_2_ subunit is binding to a unlike adhesion molecule on the plasma membrane of CHO fibroblasts, mixing these two cell types will not affect the relative amount of cell aggregates. Thus, we performed an aggregation assay combined with fluorescence-activated cell sorting (FACS). The cytoplasm of CHO wt cells was pre-stained with CMTMR and, therefore, could be detected as red when analyzed by fluorescence microscopy or FACS. The aggregation assay in Figure 5 has been carried out as described in Figure 4. In addition to pure CHO wt and pure CHO β_2_ YFP cells, we also used mixed suspensions of these two types of cells. The aggregates formed after three hours of agitation were sampled and imaged by confocal microscopy (Figure 5A). While CHO wt cells (in red) produce very small if any aggregates, the β_2_ YFP expressing cells form large ones. When mixing the two cell types in a 1:1 ratio, it can be observed that there are very few wt cells intercalated in those big (green) aggregates. Most of the wt cells (red) in the mixture were single cells. The same cell suspension was sorted simultaneously for size, granularity, and red and yellow fluorescence. The obtained data were analyzed as in Figure 4. The population of particles sorted as R2 (Figure 4B) was then sorted for red, green, and mixed fluorescence (Figure 5C). The graph in Figure 5D depicts the data obtained from three independent experiments. It can be seen that gradual dilution of β_2_ YFP expressing cells with CHO wt cells decreased the relative amount of cell aggregates proportionally. The FACS data suggest that the formed aggregates were mainly of CHO β_2_ YFP. The proportion of mixed aggregates (of both red and green color) is small and very similar in all dilutions (4% approximately). The fluorescence images and the FACS analysis shows that the aggregates are mainly formed by cells expressing β_2_ YFP, therefore suggesting that the observed cell–cell adhesion and aggregation is mediated by the β_2_ YFP on the surface of the transfected CHO cells.

### 2.5. Cell Aggregation of CHO β_2_ YFP Is Due to β_2_–β_2_ Interactions In Vivo

We have shown that CHO dog β_1_ cells form aggregates in previous work. We demonstrated that the adhesiveness is due to a direct β_1_–β_1_ interaction between adjacent cells [19]. To determine whether the β_2_ subunits of Na^+^, K^+^-ATPase located on adjacent cell membranes can trans-interact and contribute to the observed cell aggregation of CHO β_2_ YFP, we implemented an in vitro Pull-down assay. As a bait, we immobilized the β_2_ subunit through the fused YFP on Sepharose beads coupled to an anti-GFP antibody. We then used as prey cell lysates that contain or do not contain β_2_ subunits and as control lysates containing β_1_ subunit. As shown in Figure 6A, none of the pull-down assays using β_2_ YFP as bait resulted in an interacting β_2_ subunit as prey. Then, we expressed a His_6_ tagged human β_2_ subunits in MDCK cells (Figure 2A) and used it as bait, immobilized on Ni^++^ beads. As prey, we used a cell extract expressing an identical β_2_ subunit tagged with YFP. As shown in Figure 6B, there is no interaction between the two identical subunits in this case either. These results indicate that the apparent β_2_–β_2_ interaction between live cells (Figure 2 and Figure 3) cannot be reproduced in vitro by a pull-down assay, at least not under these specific experimental conditions. Therefore, we decided to try an in vivo assay of protein–protein interaction, which we named co-affinity precipitation (Co-AP) and described in detail in Materials and Methods. As shown in Figure 6C, a co-culture of MDCK cells expressing YFP tagged and His_6_ tagged β_2_ subunits form a continuous monolayer and localize the Na-pump expressing the β_2_ subunit at both homotypic and heterotypic cell–cell contacts. A cell extract of co-cultured cells expressing the recombinant human β_2_ subunits was incubated with Ni^++^-NTA beads. After washing and elution, both β_2_ subunits were detected in western blot (Figure 6D), suggesting that these two proteins interacted in vivo, throughout the co-culture.

## 3. Discussion

The β_2_ subunit of Na^+^, K^+^-ATPase has two different functions: it is an adhesion molecule on glia involved in the interaction between neural and glial cells; and a subunit of the Na^+^, K^+^-ATPase, essential for establishing a functional ion pump. In the present work, we have been focused on the adhesive function of the β_2_ isoform. We studied the expression of the human β_2_ isoform of the Na^+^, K^+^-ATPase transfected in CHO fibroblasts (CHO β_2_ YFP) and MDCK epithelial cells. The expression of the recombinant protein was observed mainly at the plasma membrane. Our significant results include increased aggregation of CHO β_2_ YFP fibroblasts that depend on N-glycosylation and a presence of trans-interaction with homotypic β_2_ subunit in vivo and in silico.

Numerous investigations have shown the adhesion/recognition functions of β_2_/AMOG. Initially, β_2_/AMOG has been shown to promote neurite outgrowth [12]. Later, liposomes containing reconstituted β_2_/AMOG adhered to neurons in vitro and a specific antibody to β_2_/AMOG inhibited this adhesion [13]. The first question was whether these functions of β_2_/AMOG depend on the pump activity. Additional studies have demonstrated that inhibition of the pump by ouabain did not affect neuron–glial adhesion [14]. Moreover, fixed and nonviable cells expressing β_2_/AMOG can still promote neurite outgrowth [20]. The clear conclusion was that there must be a receptor molecule on neurons that recognizes the β_2_/AMOG on the astrocytes surface. Early studies aimed to identify the neuronal receptor of β_2_/AMOG were focused on the fact that β_2_/AMOG but not β_1_ is responsible for that function. Since the main difference between β_1_ and β_2_ isoforms is their level of glycosylation, it was postulated that the adhesive function resides on the carbohydrates of β_2_/AMOG. The monoclonal AMOG antibody that blocks adhesion recognizes a non-glycosylated protein [12], suggesting that the carbohydrates are not responsible for the adhesion. However, co-culture experiments for neurite outgrowth on L-cells transfected with β_2_/AMOG show an increase in neurite outgrowth that was partially competed with the soluble, extracellular domain of β_2_/AMOG (recAMOG) but the non-glycosylated recAMOG had no effect [20]. Moreover, as β_2_/AMOG bears oligo-mannoses and carbohydrate epitopes (L3 and L4) common for neuronal cell adhesion molecules, it was proposed that β_2_/AMOG has a lectin activity. Nevertheless, no interaction has been detected between β_2_/AMOG and neuronal CAMs such as L1 or NCAM [13], and no lectin activity has yet been demonstrated for β_2_/AMOG.

### 3.1. The Role of N-Glycosylation

In the present work, the aggregation assay of CHO β_2_ YFP cells reveals the adhesive property of the recombinant cells, suggesting a homophilic interaction between the β_2_ subunits of Na^+^, K^+^-ATPase on the surface of neighboring cells. These results corroborate the observations previously made in transfected L-fibroblasts [20] and U87-MG glioma cells [30]. Experiments using Tunicamycin show that the N-glycosylation of β_2_ plays an important role in β_2_-mediated cell–cell adhesion. In fact, the significance of the N-glycosylation sites of β_2_ -subunit of Na^+^, K^+^-ATPase have been already demonstrated for the adhesion of Basigin/CD147 to the pump [25]. However, that interaction has been shown to occur in astrocytes, in cis, namely in the same plasma membrane. The group of Vagin, have published series of works focused on the roles of the N-glycosylation sites of the β subunits of Na^+^, K^+^-ATPase and H^+^,K^+^-ATPase [10,11,23,24,31]. As they summarize in [11], “Individual N-glycans linked to the β subunits of the Na^+^, K^+^-ATPase and H^+^,K^+^-ATPase are important for stable membrane integration of their respective α subunits, folding, stability, subunit assembly, and enzymatic activity of the pumps. They are also essential for the quality control of unassembled β subunits that results in either the exit of the subunits from the ER or their ER retention and subsequent degradation”. Here, we show (Figure 4A) that CHO β_2_ YFP cells incubated with Tunicamycin rarely express the recombinant protein in their plasma membrane, in comparison to the non-treated cells. Therefore, it suggests that N-glycosylation of β_2_ subunit of Na^+^, K^+^-ATPase is also important for the exit from ER and arrival of the Na^+^, K^+^-ATPase to the plasma membrane and thus for accomplishing cell–cell adhesion in CHO cells. Nonetheless, we cannot exclude the participation of a distinct glycosylated molecule because Tunicamycin treatment would also inhibit the N-glycosylation of other membrane proteins. Therefore, more specific experiments should be conducted. Our future studies will evaluate the in silico participation of the seven N-glycosylation sites in β_2_–β_2_ interaction, and based on that information we will choose to mutate just those sites that are suggested to be involved. If cells expressing these mutants fail to make aggregates or are not pulled down in a co-culture assay with wild type β_2_ subunit, we will be able to confirm and even to indicate which of the seven N-glycosylation sites are important for β_2_–β_2_ trans-interaction.

### 3.2. β2/AMOG Is a Heterophilic Adhesion Molecule in Astrocytes

Early study from Melita Schachner laboratory in which they generated KO mice of β_2_/AMOG suggested that β_2_/AMOG in astrocytes is mainly necessary for the proper activity of the Na^+^, K^+^-ATPase [32]. Our studies [6] and others [7] related to the Na^+^, K^+^-ATPase in epithelia [33] and in cardiomyocytes [34] revealed that Na^+^, K^+^-ATPase is not just a pump. Na^+^, K^+^-ATPase plays an important role in cell-signaling mechanisms and in cell–cell adhesion through β–β interaction. Moreover, recent works report the participation of the β_2_/AMOG in cell signaling mechanisms in astrocytes [35] and in cerebellar granule neurons [36]. β_2_/AMOG-mediated Akt phosphorylation specifically activates the mTOR/p70S6 kinase pathway implicated in cell size regulation. Consistent with that, we noticed that CHO wt and CHO β_1_ scattered cells are small and rounded particles, in comparison with the CHO β_2_ YFP scattered cells that are bigger and with irregular form. In addition, some of the CHO β_2_ YFP cells present membrane protrusions or blebs (Figure 3). Therefore, we consider that the Na^+^, K^+^-ATPase composed of β_2_/AMOG subunit participates in both ion pumping and adhesion-based cell-signaling in astrocytes. All former studies have suggested that β_2_/AMOG interacts with a heterophilic binding partner on the neuronal cell surface. Our present work does not argue with that conclusion, and actually, we are trying to identify this heterophilic molecule in our lab.

### 3.3. β2/AMOG Is a Homophilic Cell-Adhesion Molecule

Based on their protein structures in the Protein Data Bank, all the β subunits of Na^+^, K^+^-ATPase and H^+^, K^+^-ATPase are classified as a subgroup of the IgG superfamily (CATH) given their β-sandwich domain. Since β_1_ subunits are homophilic cell-adhesion molecules, we wondered about the capability of β_2_ subunits to recognize and interact with homophilic β_2_ subunits in neighboring cells. Therefore, we first evaluated this possibility using in silico approach, modeling the 3D structure of the human β_2_ subunit and applying docking analysis. Our results, showing a stable interacting interface between two extracellular domains of β_2_ subunits, encouraged us to explore this possibility further. Antonicek and Schachner [13] have already treated whether β_2_/AMOG is a homophilic adhesion molecule. They pre-incubated cerebellar neuron monolayers with an anti-AMOG antibody before the addition of AMOG-containing liposomes. They found that the antibody did not reduce AMOG-liposomes binding to neurons. Therefore, they concluded that β_2_/AMOG does not interact with itself. Our protein–protein interaction experiments in vitro neither show a homophilic interaction between β_2_ subunits. The pull-down experiments performed in this study (Figure 6A,B) suggest that the β_2_ subunits of Na^+^, K^+^-ATPase would not be able to adhere to each other in vitro, out of a cell context. One possibility to explain it could be that another factor is involved in the interaction of the β_2_ subunits between cells and that we lost it during the in vitro assay. The second possibility could be that protein folding is affected during in vitro pull-down, and as a result there is a change in the interaction site between β_2_ subunits, turning it inaccessible. On the other hand, our experiments in vivo show that cultivating CHO cells expressing human β_2_/AMOG in their plasma membrane results in adhesive cells (Figure 3). Moreover, the aggregation and adhesion assays performed with CHO β_2_ YFP cells (Figure 4 and Figure 5) and the co-affinity precipitation assay (Figure 6D) validate the homophilic property of cell–cell adhesion mediated by β_2_ subunits. Overall, previous and present findings indicate that the trans-homophilic interaction of β_2_ subunits cannot occur out of the plasma-membrane context of the cell. Since experiments evaluating the effect of anti-AMOG antibodies on astrocyte–astrocyte adhesion have shown that there is not such an interaction [13], a relevant question would be: why do astrocytes not adhere to each other by β_2_/AMOG subunits? There are several possible answers to that mystery. One can be related to recent findings reporting that in astrocytes, the Na^+^, K^+^-ATPase composed of α_2_/β_2_ is part of a protein complex formed by PrP, GluR2, α_2_/β_2_-ATPase, basigin, and MCT1 that regulates lactate transport of astrocytes and may be functional in the metabolic cross-talk between astrocytes and neurons [22]. Interestingly, basigin and β_2_/AMOG–two components of that complex–are attached to each other by a lectin–oligomannosidic interaction [25]. Therefore, this interaction could be interfering with a homophilic interaction between the β_2_ subunits of adjacent astrocytes. On the other hand, neurons do not express the β_2_ subunit, thus a homophilic cell–cell interaction between β_2_ subunits would not be responsible for the described neuron–astrocyte adhesion. Therefore, we cannot exclude the existence of a receptor molecule at the neuron surface that recognizes and binds to β_2_/AMOG. It is well established that cell adhesion molecules interacting with extracellular matrix (ECM) components regulate and modulate the formation and maintenance of synapses in the nervous system (for a recent review, see [37]). Thus, an unexplored possibility could be an interaction of β_2_/AMOG with ECM.

Based on the new findings reported here, we infer that β_2_/AMOG is a heterophilic astrocyte-neuron adhesion molecule that developed a self-avoidance mechanism in a similar way to how neurons avoid forming synapses with themselves [38]. Interestingly, recent studies have related the β_2_/AMOG with glioblastoma (GB), the most devastating type of glioma [21,30,39,40,41]. Those studies depict the β_2_/AMOG as a tumor suppressor cell adhesion molecule in astrocytes. In light of these findings, identifying an extracellular ligand for β_2_/AMOG, whether on neuron surface or at the surrounding ECM, would be of great therapeutic interest. Our great challenges in this field are understanding the self-avoiding mechanism of β_2_–β_2_ binding in astrocytes and identifying the heterophilic partner for β_2_/AMOG in Neurons.

## 4. Materials and Methods

### 4.1. Cell Culture, Transfection, and Establishment of Stable Cell Line

MDCK dog kidney epithelial cells (ATCC CCL-34) were cultured in DMEM medium (Life Technologies, Grand Island, NE, USA). CHO-K1 Chinese hamster ovary cells (ATCC CCL-61) were cultured in a mixture of F12/DMEM media. All media were supplemented with 10% fetal Calf or Bovine serum (SFB), 100 U/mL penicillin, and 100 μg/mL streptomycin under a humidified atmosphere of 5% CO_2_ and 95% air at 37 °C. The cDNAs of human β_2_ and dog β_1_ subunits of Na^+^, K^+^-ATPase cloned into pEYFP [10] were a generous gift from Dr. Olga Vagin (Department of Physiology, School of Medicine, UCLA and Veterans Administration, Los Angeles, CA, USA). In these constructs, the YFP is fused to the N-terminal (cytoplasmic tail) of the β subunits of Na^+^, K^+^-ATPase. To establish CHO cells stably expressing YFP-β_2_ or YFP-β_1_, cells were transfected over night with pEYFP-β_2_ and pEYF-β_1_ using Lipofectamine 2000 (Invitrogen) according to the manufacturer’s instructions. Thereafter, the transfection medium was removed, and the cells were incubated for an additional 24 h with fresh medium containing 10% SFB to allow recovery. For generating stable clones, 48 h after transfection, cells were harvested by trypsinization on 100 mm dishes in DMEM medium containing 800 μg/mL G418 (Gibco, Grand Island, NY USA). After 2–3 weeks, resistant clones were selected and re-cloned with the aid of FACS. Stable colonies were maintained in the presence of 200 μg/mL G418.

For generating His_6_ tagged human β_2_ subunit, the cDNA of β_2_ was amplified from pEYFP-β_2_ plasmid using the following primers: FORWARD (5′AGCTGAATTCGTCATCC AGAAGAGAAGAAGAGCTG-3′) and REVERS (5′ TTTTGAGCTCTCAGGTTTTG TTGATGCGGAGTTTGAAG GC-3′). The cDNA was re-cloned in pcDNA4 HisMaxB vector (at EcoRI and XhoI restriction sites) generating pCDNA β_2_-His_6_ vector. This vector was further transfected as described above in MDCK cells and stable clones were selected using G-418 and verified by IF and WB using anti β_2_ antibody (MDCK β_2_ His_6_).

### 4.2. Immunofluorescence Imaging

Immunofluorescence was performed as described [7]. Briefly, cells were washed ice-cold with phosphate-buffered saline PBS solution (CaCl_2_, 0.90 mM; MgCl_2_ anhydrous, 0.49 mM; KCl, 2.66 mM; KH_2_PO_4_, 1.47 mM; NaCl, 137.93 mM; Na_2_HPO_4_, 8.09 mM), fixed and permeabilized with methanol for 10 min at −20 °C, washed three times with PBS, then blocked with 5% bovine serum albumin BSA (Bovine Serum Albumin Cohn fraction V powder, Equitech-Bio, INC., Kerrville, TX, USA) for 1 h at RT. The cells were incubated for 1 h at 37 °C with a specified primary antibody, washed, and then incubated (30 min, RT) with a secondary antibody; anti-mouse or anti-rabbit antibody according to the primary antibody used. The secondary antibodies used in this study were conjugated to Alexa 488, Alexa 594, or Cy5. When assaying for β_2_-YFP expression, no primary or secondary antibodies were used and fluorescence was observed at 530 nm. Coverslips were washed, fixed with methanol, and mounted for observation and imaging using confocal microscopy (SP2 Leica Microsystems). Captured images were processed with ImageJ (National Institutes of Health).

### 4.3. Primary and Secondary Antibodies

Monoclonal antibodies against the Na^+^, K^+^-ATPase β_2_ subunit (BD), against GFP, clones 7.1 and 13.1, which also recognizes YFP (Roche Diagnostics); against the Na^+^, K^+^-ATPase β_1_ subunit donated by Dr. Caplan, M. (Yale University, New Haven, CT, USA). Polyclonal antibody against GFP, which recognizes YFP (Santa Cruz Biotechnology, Santa Cruz, CA, USA). The following secondary antibodies were used: CY5-goat anti-mouse IgG (Life Technologies, Eugene, OR, USA) and for western blot, AffiniPure Goat Anti-Mouse IgG, light chain specific conjugated to horseradish peroxidase (Jackson Immuno Research Labs. Baltimore Pike., West Grove, PA, USA).

### 4.4. Cell Aggregation Assay

Cell aggregation assay was performed as described [6]. Briefly, cells were washed with PBS, incubated with trypsin 0.05% and verseno 0.05% at 37 °C for 5 min, and dispersed by gentle pipetting. Next, 1 × 10^6^ cells were resuspended in P buffer (145 mM NaCl, 10 mM HEPES, pH 7.4, 1.0 mM Na-pyruvate, 10 mM glucose, 3.0 mM CaCl_2_) and complemented with protease inhibitor mix (GE Healthcare Bio-Science Corp., Piscataway, NJ, USA). The cell suspension was placed in 1.5-mL microfuge and rotated on a gyratory shaker at 37 °C for 3 or 15 h as indicated in each figure. Aggregation was stopped by adding 2% (*vol*/*vol*) glutaraldehyde. The extent of aggregation was assessed by flow cytometry analysis (fluorescence-activated cell sorting: FACS Vantage; BD Biosciences, San Jose, CA, USA). Then, 10,000 events were sorted for size and granularity and plotted. The dead cells population was eliminated and the proportion of big and granulated particles out of the live particles was calculated. Cell treatment with Tunicamycin previous to aggregation assay was as follows: Confluent monolayers were incubated for 24 h without or with 1 µg/mL of Tunicamycin. Afterward, cells were washed three times with PBS and incubated for one hour in a culture medium. Aggregation assay proceeded as described above.

### 4.5. Cell Adhesion Assay (Dispase Assay)

The Dispase-based dissociation assay reported by [29] was adapted to CHO fibroblast as described by [42]. Cell cultures were seeded onto 24-well dishes. Twenty-four hours after reaching confluency, cultures were washed three times in phosphate-buffered saline (PBS) and incubated in the same for 30 min. Thereafter, the cells were incubated in 200 μL of Dispase I (0.6 U/mL; Sigma, St. Louis, MO, USA) for 40 min. The Dispase solution was carefully removed and 500 μL of culture medium (DMEM/F12 1:1 without complement) was added. Afterward, five rounds of pipetting through a 1 mL Gilson pipette were conducted. The dissociated cells and fragments were then sampled by taking three drops of 30 µL each and depositing on a slide. Fragments were imaged using an inverted microscope (Axiovert 200 M, Carl Zeiss, Oberkochen, Germany).

### 4.6. Mixed Monolayers (Co-Cultures)

Cell mixture was performed as described previously [5,43]. Briefly, the cell type different from β_2_ YFP transfected CHO fibroblast (CHO wt or MDCK β_2_ –His_6_) was incubated with CellTracker Orange CMTMR (Molecular Probes, Eugene, OR, USA) for 1 h on 37 °C at a final concentration of 6.0 μM. Cells were then washed three times with PBS solution, re-incubated for 1 h in DMEM supplemented with 10% FCS or FBS. The day after, cells were harvested by trypsin and the suspension was mixed with CHO β_2_ YFP cells suspension, in different proportions, as indicated in each case, to be co-cultured on glass coverslips and processed the day after for immunofluorescence assay. The same procedure was implemented when the cell mixture was analyzed for cell aggregation except for when the mixing ratio was 3:1, 1:1, and 1:3. For Co-AP assay, MDCK β_2_-His_6_ cells were co-cultured with CHO β_2_ YFP cells or with MDCK β_2_ YFP cells in a 3:1 ratio without using a CellTracker.

### 4.7. Pull-Down Assay

Pull-down assays were conducted in two ways: (1) MDCK cells stably expressing the human β_2_-His_6_ were cultured for 48 h. Thereafter, lysed with RIPA buffer (Santa Cruz Biotechnology, Santa Cruz, CA, USA) containing protease inhibitor mix (GE Healthcare Bio-Science Corp., Piscataway, NJ, USA), and protein content was determined with BCA Protein Assay (Thermo-Scientific Meridian Rd., Rockford, IL, USA). β_2_ His_6_ as bait was immobilized on Ni-NTA beads (His Trap FF column, GE Healthcare, Chicago, IL, USA) previously equilibrated with 10 mL RIPA containing protease inhibitors mix. Generally, 2 mg of total cell extract protein were loaded and allowed to interact overnight, at 4 °C, with gentle shaking. After 10 washes with 10 mL of 5 mM imidazole and 10 washes with 10 mL of 10 mM imidazole the lysates of CHO β_2_-YFP and MDCK β_2_-YFP cells were loaded as prey and interaction were allowed to occur overnight at 4 °C. After washing with 5 mM and 10 mM imidazole as above, the protein complexes were eluted using a solution of 500 mM imidazole. The eluted samples were mixed with loading buffer, heated at 90 °C for 5 min, then loaded on a 10% SDS-PAGE and analyzed by Western blot using anti-YFP antibody or anti-Na^+^, K^+^-ATPase β_2_ antibodies. (2) CHO YFP-β_2_ monolayers were cultured for 48 h and then lysed as described above. In this method, a YFP-tagged protein is immobilized by GFP sepharose beads and acts as the bait to capture a putative binding partner (prey). Here, lysates of ARPE-19 cells and of rat brain were loaded as prey, and interaction was allowed to occur overnight at 4 °C. The protein complexes are eluted boiling with loading buffer 2× for 5 min at 90 °C and then subjected to Western blot analysis as specified above.

### 4.8. Co-Affinity Precipitation (Co-AP)

MDCK and CHO cell lines, both stably expressing the human β_2_ subunit, were cultured as described above. Next, 24 h after seeding, both cell lines were harvested by trypsinization and seeded in a 100 mm petri dish, 30% were MDCK β_2_-His_6_ and 70% CHO β_2_ YFP or MDCK β_2_ YFP, and cultivated in an F12/DMEM mixture supplemented with 10% FBS and 100 U/mL of penicillin, 100 ug/mL of streptomycin. After 48 h, the cells were lysed with RIPA buffer (Santa Cruz Biotechnology, Santa Cruz, CA, USA) containing protease inhibitors (GE Healthcare Bio-Science Corporation, Piscataway, NJ, USA). The protein content was determined with BCA Protein Assay (Thermo-Scientific Meridian Rd., Rockford, IL, USA) 2 mg of total proteins from cell extract was placed in a 1.5 mL tube with 100 μL of HIS-Select^®^ HF Nickel Affinity Gel (Sigma H0537-25ML) and were incubated overnight at 4 °C with a gentle shaking. The next day, three washes with 250 μL of cold PBS, three washes with 250 μL of 5 mM imidazole, and an additional three washes with 10 mM imidazole were carried out. Finally, the protein was eluted with 200 μL of 500 mM imidazole (stirring for 10 min at 4 °C), centrifuged, and the eluate was mixed with 2× loading buffer and heated for 5 min at 90 °C. The whole sample was run on an 10% SDS-PAGE and analyzed by Western blot using an anti-β_2_ antibody.

### 4.9. Molecular Modeling

The three-dimensional (3D) structure of the human Na^+^, K^+^-ATPase β_2_ (ATP1B2; Uniprot ID: P14415) was obtained by employing the Swiss Model Program [26]. For the In Silico studies, only the extracellular domain of the ATP1Β2 was considered (residues 69 to 289). The 3D model of the extracellular domain (EDβ_2_) was built by contemplating the crystal structure of ATP4B of Sus scrofa (PDB ID: 5YLU, (Chain B) showing an identity of 39.82%. The 3D model includes the seven N-glycosylation sites: Asn96, Asn118, Asn153, Asn159, Asn193, Asn197, and Asn238, and the three conserved disulfide bridges: Cys129-Cys150, Cys160-Cys177, Cys200-Cys261, which were built-in by using the CHARMM-GUI Program [44].

### 4.10. Molecular Docking Studies

We performed docking of EDβ_2_ (Chain A)-EDβ_2_ (Chain B) by using the HDOCK Server [27,28]; different dimer complexes were obtained and the most energetically favorable dimer was chosen. Analysis of the interactions in the interface of the dimeric structure of EDβ_2_ was obtained employing the PDBsum web server [45]. UCSF Chimera Software was employed to get the images of the monomeric and dimeric structures of EDβ_2_ [46]. Ramachandran plot was obtained by using SAVES 6.v.0 Software [47].

### 4.11. Statistical Analysis

This was performed using a chi-square test (GraphPad Prism 4 and Microsoft Excel). Statistical significance is specified in the figure legend. Error bars indicate standard deviation.

## 5. Conclusions

β_2_/AMOG has been known as a cell adhesion molecule for more than 30 years. So far, the known interactions of β_2_/AMOG expressed in astrocytes are heterophilic cis-interactions. A trans-heterophilic interaction with a yet unknown neuronal molecule is also strongly suggested. In the present work, we show that β_2_/AMOG is a homophilic adhesion molecule when expressed on the plasma membrane of non-astrocytic cells such as fibroblasts (CHO) or epithelial cells (MDCK). Recent findings indicating the participation of β_2_/AMOG in glioblastoma pathology makes it an interesting target for therapeutic approach. Our future studies will focus on characterizing the molecular mechanism for β_2_–β_2_ homophilic interaction and the role of N-glycosylation in that mechanism.

## Figures and Tables

**Figure 1 ijms-23-07753-f001:**
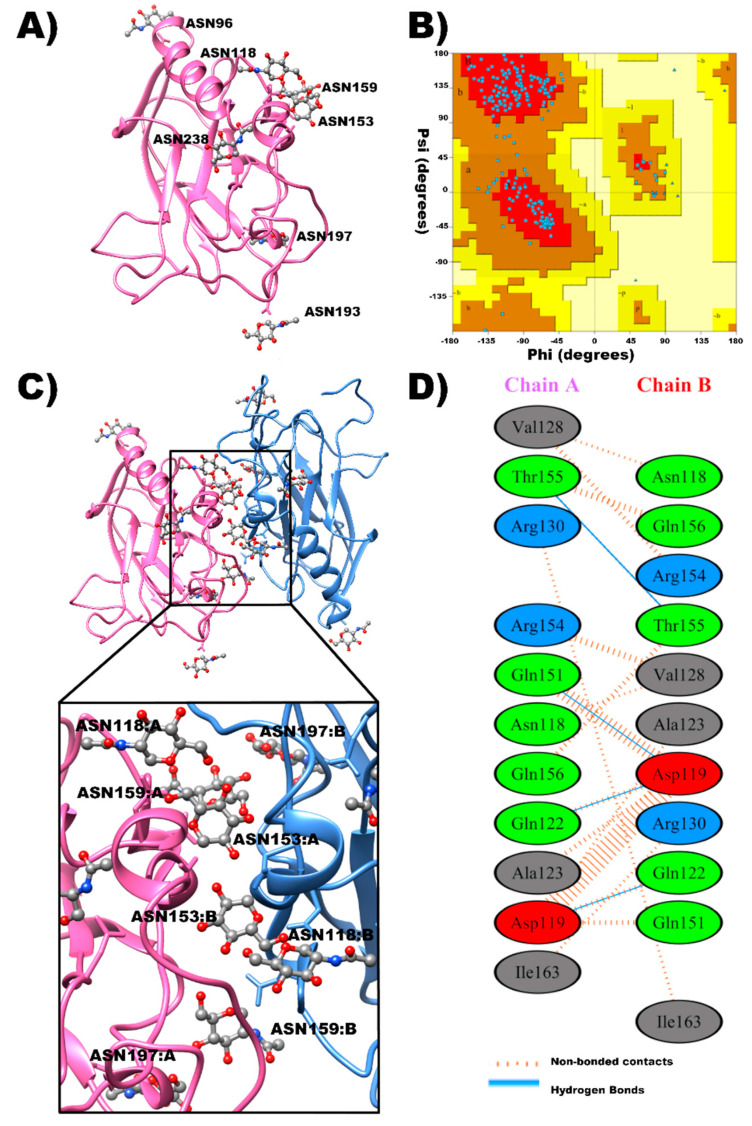
**Monomeric and dimeric 3D structures of EDβ_2_**. (**A**) 3D structure of the EDβ_2_ monomer including its 7 N-glycosylation sites. (**B**) Ramachandran plot of the EDβ_2_ monomer. (**C**) Dimeric structure of EDβ_2_, in which Chain A is colored in pink, Chain B is colored in blue, and N- glycosylation sites are marked in ball and sticks. An expanded view of the putative trans binding interface between adjacent EDβ_2_ is illustrated to visualize the four N-glycosylation sites. (**D**) Residue–residue interactions in the interface of the dimeric structure of EDβ_2_.

**Figure 2 ijms-23-07753-f002:**
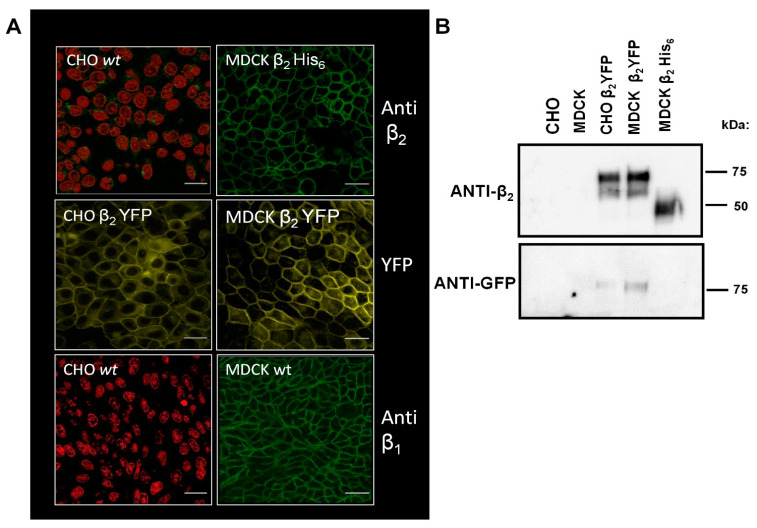
**Generation and description of CHO and MDCK cells expressing the β_2_ subunit**. (**A**) Stable cell lines expressing the β_2_ subunit, tagged with the Yellow fluorescent protein (β_2_ YFP) and with 6xHis tag (β_2_ His_6_) were created from CHO-K1 cell line (CHO β_2_ YFP) and from MDCK cell line (MDCK β_2_ His_6_ and MDCK β_2_ YFP). Immunofluorescence images of wild type CHO cells and MDCK β_2_ His_6_ cells assayed for β_2_ expression by a specific anti-β_2_ antibody are shown in upper panels. Fluorescent images of β_2_ YFP expressed in CHO and MDCK cells are shown in middle panels. Immunofluorescent images of β_1_ subunit expression assayed with a specific anti-β_1_ antibody in wild type CHO and MDCK cells are shown in lower panels. Nuclei are stained with propidium iodide in red (bar scale 50 μm). (**B**) Western blot analysis of Wild type and transfected cells (as indicated in each lane) for the expression of β_2_ subunit (upper panel) and for YFP expression (lower panel).

**Figure 3 ijms-23-07753-f003:**
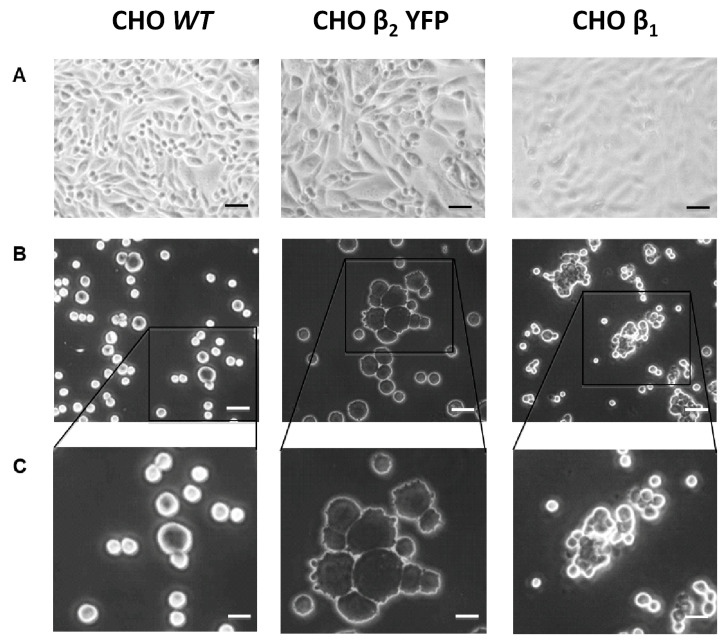
**CHO cells expressing the β_2_ YFP subunit are highly adhesive.** (**A**) A representative bright field image is shown for the indicated CHO fibroblast taken before Dispase I treatment. (**B**) Following 40 min of incubation with dispase I, the cells were subjected to mechanical force by pipetting and the dissociated particles were dropped on slides (30 μL) and imaged in bright field with 20× objective. (**C**) Larger image of cell particles of each cell line is shown from the inset in panel B (bar scale in A and B is 30 μm, in C 10 μm).

**Figure 4 ijms-23-07753-f004:**
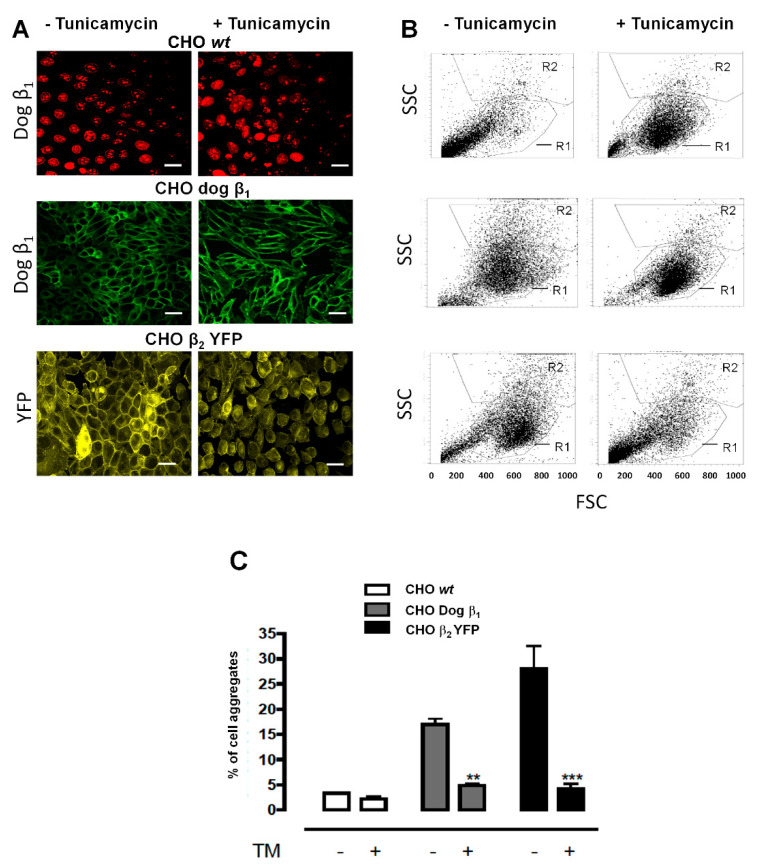
**Cell aggregation of CHO cells expressing β_2_ YFP depends on the presence of N-glycan.** (**A**) The indicated CHO cells were cultured in the absence and presence of Tunicamycin as described in Methods. A confocal image of each cell type during culture is shown. CHO wt cells (Upper panels) and expressing dog β_1_ subunit (Middle panel) are treated for immunofluorescence with anti-β_1_ antibody. CHO β_2_ YFP cells (Lower panel) are imaged for YFP auto-fluorescence. (**B**) The percentage of cell aggregates formed was calculated by flow cytometry. A representative dot plot of Forward Scatter (FSC) versus Side Scatter (SSC) is presented for each cell type. The percent of events in R2 represents the percent of cell aggregates. (**C**) The percent of cell aggregates in R2 were averaged and illustrated in a comparative bar graph. The amount of cell aggregates of fully glycosylated CHO dog β_1_ and CHO β_2_ YFP cells is significantly higher than that of CHO wt cells. Error bars, ±S.D. (*n* = 5); **, significant difference from CHO dog β_1_, *p* < 0.005, ***, significant difference from CHO β_2_ YFP, *p* < 0.005.

**Figure 5 ijms-23-07753-f005:**
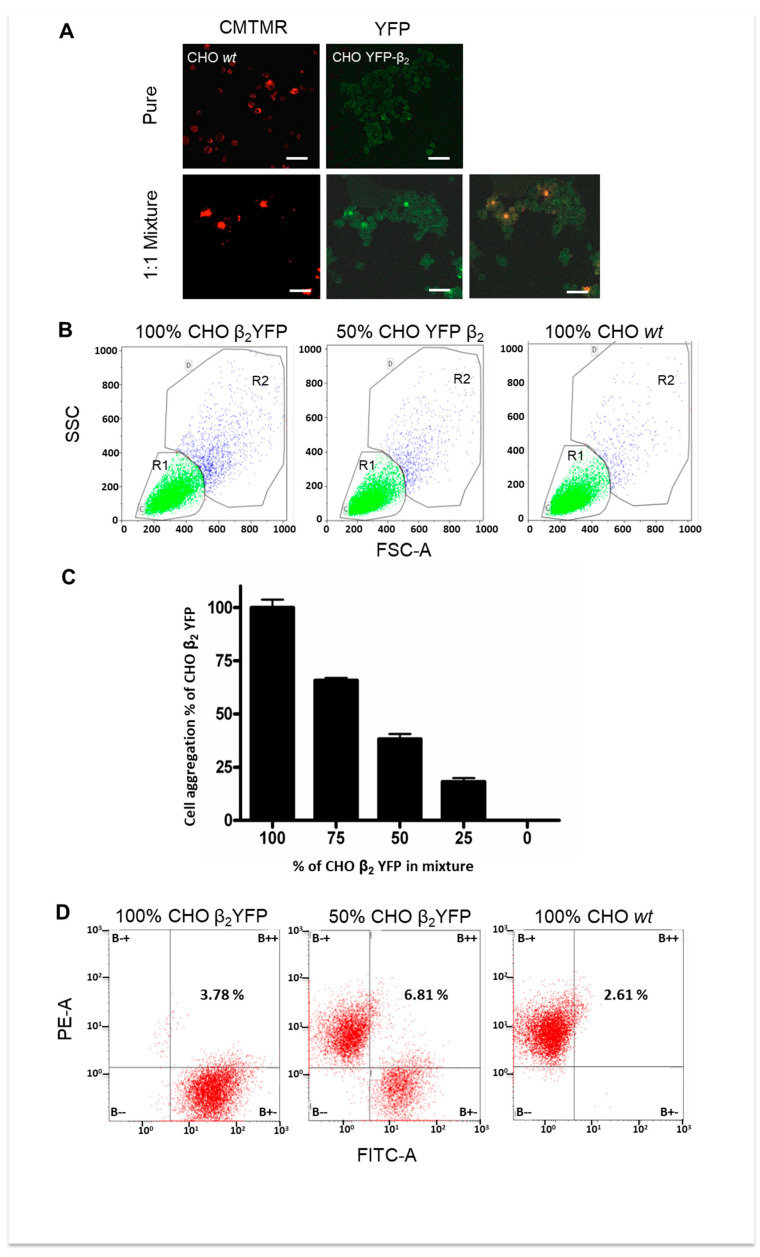
**Mixing CHO β_2_ YFP with CHO wt cells decreases the extent of cell aggregation.** (**A**) CHO wt and CHO β_2_ YFP cells were cultured for cell-aggregation assay as in Figure 3. CHO wt were pre-stained with CMTMR. In addition to the pure cell lines, three mixtures of CHO β_2_ YFP cells and CHO wt have been assayed. A 30 μL sample of each tube was dropped on slides and imaged by a confocal microscope. The upper panel is for aggregates of pure CHO wt cells imaged in red and pure CHO β_2_ YFP cells imaged in green. The lower panel shows aggregates formed by a 1:1 mixture of CHO wt and CHO β_2_ YFP cells imaged in red and green. (Bar scale is 30 μm). (**B**) Representative dot-plots from flow cytometry assay are shown. The events in R2 are the aggregated population. (**C**) The aggregates of R2 were also analyzed by fluorescence-activated cell sorting. Representative dot-plots of red vs. green fluorescence are shown. Numbers are the proportion of mixed aggregates in percent. (**D**) The extent of cell aggregation of the three different mixtures (3:1, 1:1, and 1:3 ratios) was illustrated as the percent of the aggregation of pure CHO YFP-β_2_ cells. Values are mean ± SED from four independent experiments.

**Figure 6 ijms-23-07753-f006:**
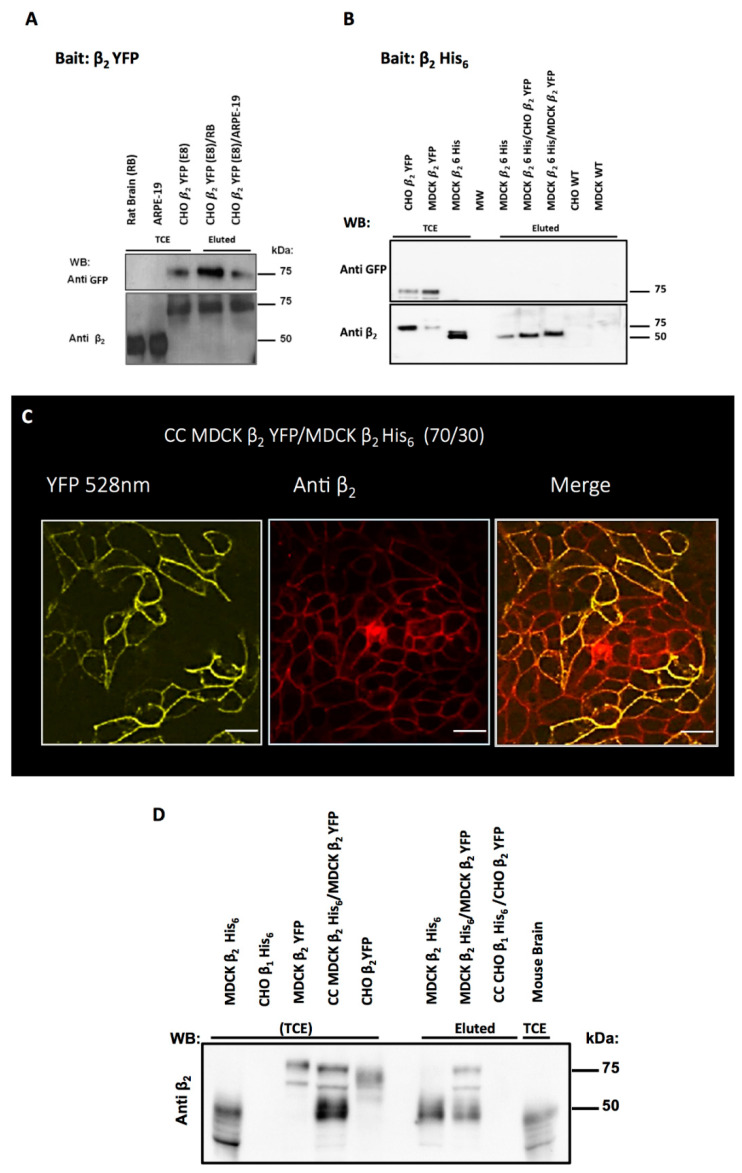
**Protein–protein interaction analyses of β_2_ subunit.** (**A**) For pull-down assay, β_2_ YFP was immobilized on GFP-sepharose beads (bait). Then, the lysate of rat brain or ARPE-19 cells expressing β_2_ subunit (prey) were incubated to form β_2_–β_2_ complexes. Western blot analysis shows the expression of β_2_ subunit in ARPE-19 cells and rat brain extract (TCE—Total Cell Extract). Both anti-GFP and anti-β_2_ antibodies detect the β_2_ subunit in CHO β_2_ YFP cells. Nevertheless, none of the assayed β_2_ subunits were pulled down by immobilized β_2_ YFP. (**B**) A pull-down assay using β_2_ His_6_ as bait, immobilized on Ni-NTA HF resin. Total extracts of CHO β_2_ YFP and MDCK β_2_ YFP cells (prey) were incubated to form β_2_–β_2_ complexes. WB analysis shows the expression of β_2_ subunits in all transfected cell lines. None of the prey proteins (tagged with YFP) appears to form β_2_–β_2_ interaction (Eluted) on the Ni-NTA HF resin. (**C**) IF images of a co-culture of MDCK cells transfected with β_2_ YFP or β_2_ -His_6,_ in a 70/30 ratio. On the left panel, an image of cells expressing β_2_ YFP are shown in Yellow (528 nm), in the middle panel, β_2_ -His_6_ was stained with anti-β_2_ antibody (Red), and on the right panel, a merged image showing continuous orange cell–cell contacts between the two cell-lines. (**D**) Co-affinity precipitation assay (Co-AP) performed with a co-culture as described in (**C**). As a negative control, a similar co-culture was made with CHO cells expressing β_1_-His_6_ and MDCK β_2_ YFP. Total cell extracts of the co-cultures were loaded on Ni-NTA HF resin to separate β_2–_β_2_ complexes that were formed in vivo. Eluted samples were analyzed by WB. Only the eluate of β_2_ -His_6_ and β_2_ -YFP co-culture show a pattern of bands corresponding to that of the cell extracts of β_2_ -YFP and β_2_ -His_6_.

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
