# Peer review of "The β2-Subunit (AMOG) of Human Na+, K+-ATPase Is a Homophilic Adhesion Molecule"

_ijms, 2022, doi:10.3390/ijms23147753_

Round 1

Reviewer 1 Report

The manuscript from Shoshani’s group titled “The beta2-subunit (AMOG) of human Na+,K+-ATPase is a homophilic adhesion molecule” describes the dimer structure model of beta2 subunit, and then it shows that CHO fibroblasts transfected with the beta2-subunit become more adhesive and make large aggregates. And Co-AP assays show the interaction between β22- β22.

However, despite the above findings, the manuscript still has some comments that need to be revised.

1. The resolution of Fig.1 and Fig.6 is not good and is difficult to see.

2. Using the HDOCK program produces β22 dimer structure model, how to decide on this model? As usually the HDOCK program always provides many models, it cannot be decided the model by the orientation of the monomers, especially in this paper also shows the interaction site. If there is some other assay to investigate the interaction site, would confirm the accuracy of this structure model.

3. In Figure S1, it seems that these figures are generated by LIGPLOT program, but it does not mention in the method part. In addition, the analysis of these figures also has some mistakes.  In Figure S1A, what is the distance between the interaction atoms of glycosylated Asn118 (A) with Gln 151 (B)? In Figure S1D, it seems all the interactions are from chain B, it is not related to the interaction between two monomers.  

4. It is strange that the interaction is shown from the live cells, but not from the pull-down assay. Because in the pull-down experiment, it also used the CHO b2-YFP. Is that possible the concentration?

5. The writing needs to be improved, especially the discussion part, it has “Nevertheless” almost in every paragraph. The abbreviated term should be shown when the author first mentions it, like CHO and MDCK in the abstract. β22/AMOG or AMOG/β2  needs to be consistent. 

Author Response

Dear Reviewer,

I am attaching a reply for your comments.

Thanks for your helpful recommendations.

Liora Shoshani

Reviewer 2 Report

The article is of research interest. The results will broaden the understanding of the role of the Na+,K+-ATPase beta subunit.

Comments:

1. It is recommended to structure the abstract.

2. It is recommended to improve the quality of figures (1 and 6). The text and images are poorly readable.

3. It is recommended that articles from the last 3 years be added to the list of references.

4. It is recommended to add a conclusion with a summary of the findings and their significance. It is also recommended to add future research perspectives.

Author Response

1. It is recommended to structure the abstract.

Reply: Thanks! We did some changes and hope it is now clearer.

2. It is recommended to improve the quality of figures (1 and 6). The text and images are poorly readable.

Reply: We agree! Converting the Word manuscript to PDF decreases the quality of the image. We have worked again on Figures 1 and 6 to get a higher resolution.

3. It is recommended that articles from the last 3 years be added to the list of references.

Reply: Thanks for your observation. References related to β2/AMOG from recent years were added and discussed (see refs: 31, 32, 34-37).

4. It is recommended to add a conclusion with a summary of the findings and their significance. It is also recommended to add future research perspectives.

Reply: Good recommendation! We now include a conclusion at the end of the text which includes also future perspectives.

Thanks for all your recommendations!!!

Reviewer 3 Report

In present manuscript authors use structure prediction to create a model of
β2 subunit (AMOG) of human Na+, K+-ATPase and validate that in cells β2 subunit could interact with neighboring cells β2 subunit. This leads to adhesion in monolayer cells and aggregation in suspension cells.
They also established using tunicamycin treatment that N-glycan modification take precipitate in the process. Further, they have performed in-vitro and in-vivo pull-down assay to validate the interaction however it shows contradictory results between in-vitro v/s in-vivo condition. This is one of the drawbacks of this study.

Following are the concern related to the manuscript.

1)    Author should discuss the possible factors could cause their contradictory results in in-vitro pull down assay. One possibility could be that during In-vivo other factor also could involve in the interaction of the β2 subunit between cells. Which could not be present during in-vitro assay.  Second possibility could be protein folding could be affected during in-vitro pull down and could lead to change in accessibility in interaction site between β2 subunits. This could lead to loss in interaction in in-vitro assay.

2)    Author have used ATP4B crystal structure as template (which is only 39.82% identical) for building the model of EDβ2. Author should use more model independent/unbiased template approach such as recently AI prediction tool AlphaFold2 (https://www.nature.com/articles/s41586-021-03819-2) for construction of β2 subunit model.

3)    Figure 2A, CHO wt panel what is red color representing?

4)    Although author have mentioned in method anti-GFP antibody is used for detection of YFP. Lane152 says figure2B lower panel is anti-YFP antibody while in figure2B it mentioned as anti-GFP antibody. It should be consistent (either anti-GFP or anti-YFP) at both places.

5)    Author have mentioned that β2 based aggregation depends on glycosylation modification and tunicamycin treatment leads to decrease in cell aggregation in CHO cells. Although the tunicamycin treatment is not specific β2 subunit and could affect glycosylation at other proteins also. Author should show the impact of glycosylation on aggregation specific to β2 subunit by site directed mutagenesis of glycosylation residue on β2 subunit and perform the co-expression pull down assay with wildtype protein. This will establish direct role of glycosylation modification in β2 subunit interaction/function.

6)    Lane 537 the software is UCSF Chimera not USF Chimera.

Author Response

Thanks for your recommendations.

Liora Shoshani

Round 2

Reviewer 1 Report

Comments:

1.       It is better to show the superimpose of models generated by the Swiss model program and Alpha fold program, although RMSD is low, sometimes it still shows the difference of some important sites (like the interaction sites). The author can show the speculated interaction sites of two models to show it is structural conservation.

2.       It is better to label the 7 N-glycosylation sites in Fig.1A. Do not use the spheres mode to show the glycosylation, it cannot show the model of the structure. It is better to use the cartoon model to show the overall structure and use stick mode to show the 7 N-glycosylation sites.

3.       In Fig.1D, the Residue-residue interactions at the interface are different from supplementary Fig.1, such as in supplementary Fig.1B the hydrogen bond is between Asn153 (A) and Asn 153 (B) does not show in Fig. 1D.

4.       It still has some questions about Supplementary Fig.1, a Van der Waals interaction can not support the formation of the dimer, it needs Hydrogen Bonds. However, in supplementary Fig.1A, the hydrogen bond is between the same molecules, Asn118 (A) and Ser 120 (A). This kind of intramolecular hydrogen bond does not help to from the dimmer.

5.       In the response letter, the author showed the Hdock models. However, the binding energy for the 9 models does not show big differences, then what is the structure difference, do they have the same interaction sites? If it shows the different interaction sites, then the author cannot use this model.

6.       Page 7, line 235. Fig.s 4B and C ?
